# Chikungunya Virus E2 Structural Protein B-Cell Epitopes Analysis

**DOI:** 10.3390/v14081839

**Published:** 2022-08-22

**Authors:** João Paulo da Cruz Silva, Marielton dos Passos Cunha, Shahab Zaki Pour, Vitor Renaux Hering, Daniel Ferreira de Lima Neto, Paolo Marinho de Andrade Zanotto

**Affiliations:** Laboratory of Molecular Evolution and Bioinformatics, Department of Microbiology, Biomedical Sciences Institute, University of São Paulo, São Paulo 05508-000, Brazil

**Keywords:** Chikungunya virus, immunoinformatics, B-cell epitopes, peptides, ELISA

## Abstract

The *Togaviridae* family comprises a large and diverse group of viruses responsible for recurrent outbreaks in humans. Within this family, the Chikungunya virus (CHIKV) is an important *Alphavirus* in terms of morbidity, mortality, and economic impact on humans in different regions of the world. The objective of this study was to perform an IgG epitope recognition of the CHIKV’s structural proteins E2 and E3 using linear synthetic peptides recognized by serum from patients in the convalescence phase of infection. The serum samples used were collected in the state of Sergipe, Brazil in 2016. Based on the results obtained using immunoinformatic predictions, synthetic B-cell peptides corresponding to the epitopes of structural proteins E2 and E3 of the CHIKV were analyzed by the indirect peptide ELISA technique. Protein E2 was the main target of the immune response, and three conserved peptides, corresponding to peptides P3 and P4 located at Domain A and P5 at the end of Domain B, were identified. The peptides P4 and P5 were the most reactive and specific among the 11 epitopes analyzed and showed potential for use in serological diagnostic trials and development and/or improvement of the Chikungunya virus diagnosis and vaccine design.

## 1. Introduction

The Chikungunya virus (CHIKV) (family: *Togaviridae*; genera: *Alphavirus*) [1,2,3] is a viral species widely distributed in the world. CHIKV outbreaks impact, in terms of morbidity and socioeconomic problems, areas with circulation [2,4,5,6]. Currently, three genotypes have been identified based on the genealogical relationships within the CHIKV: (i) East-Central-South Africa (ECSA), (ii) West African (WA), and the (iii) Asian genotype. Within the ECSA genotype, there is a lineage that is considered very important due to its biological characteristics: the Indian Ocean Lineage (IOL). CHIKV outbreaks are aggravated by a simultaneous co-circulation with other arboviruses, including the Dengue, Zika, and Mayaro viruses, which share symptoms in acute symptomatic individuals, which is a limitation for the clinical diagnosis and treatment [6,7,8,9,10].

The development of methods for synthetic immunogen synthesis has become increasingly applied and explored in combination with the development and improvement of bioinformatics algorithms for predicting immunogenic targets and identifying potential immune molecules of interest [11,12,13,14,15,16]. Epitopes can be linear or conformational, being linear arranged in a protein (the sequence of amino acids of the primary structure), while the conformational are discontinuous in the amino acids sequence and are linked to the secondary, tertiary, and quaternary structures of the proteins. Specific antibodies secreted by B lymphocytes can recognize and bind to both the linear and conformational epitopes of the antigens [11,12,13,17,18]. The mapping epitopes technique has numerous and important applications, including the development of new vaccines and the improvement of existing vaccines. This process is fast, efficient, and key to understanding the immune system for any type of protein [11,12,14,15,18,19]. Antigenic recognition is based on determining the recognized CHIKV immunogenic proteins by specific antibodies generated in the convalescent phase of infection. During the CHIKV outbreak in Singapore, occurring in 2008, it was observed that response antibodies against most of these protein determinants were elevated in 2 to 3 months but progressively decreased, as occurs in other infections. The only response against E2 glycoprotein was still detectable at 21 months post-infection. This long-standing response against the N-term region of glycoprotein E2 (the amino acids’ positions between 1 and 18) makes it an important candidate for specific serological diagnostic tools [12,13,19].

There have been no studies on the specific targets of the anti-CHIKV antibody-mediated immune response of infected patients in Brazil. At the moment, there is no approved vaccine, and no effective antiviral agents have been made available so far. Treatment for the virus infection is often limited to symptomatic treatment due to problems in drug specificity and effectiveness [20,21,22,23,24]. However, recent epidemiological data reveal increasing evidence of the importance of antibody-mediated protection against the CHIKV, where it is important to highlight the possibility of using anti-CHIKV antibodies in therapeutic or prophylactic treatment [12,13,19,20,25].

Within this context, the epitopes’ validation promotes a greater understanding of the interaction of peptides and the activation of the immune response. Together, these observations strongly imply the importance of the interactions between the specific antibodies and the peptides corresponding to the amino acid level. To improve key information in the development of vaccines and diagnostic tests for the CHIKV, considering specific amino acid residues important in the recognition of anti-CHIKV antibodies [11,12,13,19,20], we aim to: (i) map the epitopes of the CHIKV structural glycoproteins (E2 and E3) to identify specific IgG class antibody binding sites of the CHIKV’ previously infected patients and identify the reactive epitopes of anti-CHIKV IgG antibodies induced by primary infection in humans infected with the CHIKV ECSA genotype; (ii) identify antigenic signatures that may be used in the development of serological diagnostic trials and/or vaccine development. While this topic has already been addressed by researchers in other regions, in this study, we aim to bring a novel perspective on the still underexplored topics of CHIKV infection in South America [3,6,12,13,16,19,26].

## 2. Material and Methods

### 2.1. Ethics Committee

The serum samples for the present study were collected in Sergipe [27] following a protocol approved by the Research Ethics Committee of the Institute of Biomedical Sciences of the University of São Paulo (protocol CEPSH-ICB nº 1406/17). All consenting adults signed a written informed form, and for children, the signature was made by the parents or guardians with the written informed consent on their behalf. These samples were collected during a period of intense circulation of the Chikungunya virus in the northeast region of Brazil [27,28,29].

### 2.2. Phylogenetic Reconstruction

To choose representative sequences of all CHIKV genotypes and lineages, we performed an analysis of the phylogenetic reconstruction using 63 sequences previously analyzed by our group [28,30] and available in Genbank (http://www.ncbi.nlm.nih.gov) (accessed on 1 October 2020), referring to the complete genomes representing all genotypes. The phylogeny was performed with the maximum likelihood analysis using IQ-TREE software (James Barbetti, Camberra, Australia) (http://www.iqtree.org/) (accessed on 1 October 2020) using the standard criteria [31]. All sequences of this study are presented in the tree in the format: genotype/access number/country of isolation/year of isolation [31]. To predict the antigenicity and linear and conformational epitopes, we selected the Chikungunya virus’ complete genome sequences representing the four CHIKV genotypes (Asian, ECSA, IOL, and WA), which were used in the computational modeling as described in the next section (GenBank accession numbers: KP164572, KY055011, HM045817, and KJ796852).

### 2.3. Computational Modeling

Structural protein modeling of the CHIKV was performed using the homology strategy based on crystallographic structures available at the Protein Data Bank (PDB: 3N43) and the Chikungunya virus’ complete genome sequences (GenBank accession numbers: KP164572, KY055011, HM045817, and KJ796852). These genome sequences were translated into amino acid sequences, and the corresponding regions of structural proteins E3 and E2 were separated and modeled individually and validated. This step was performed using I-TASSER v5.1 software, using the default settings and the gnu parallel option. Furthermore, the species that had their structures identified in crystals were aligned to our models, and the RMSD calculations were made [32]. The models were validated using the MolProbity website for flips and stereochemical corrections, and on the ProSA website, Ramachandram plots were generated to validate the models [33,34]. Using the obtained structures in the modeling, the antigenicity and conformational and linear epitopes were predicted. The antigenicity score was obtained by the Kolaskar and Tongaonkar antigenicity scale, a semi-empirical method that makes use of the physicochemical properties of amino acid residues and their frequencies of occurrence in experimentally known segmental epitopes were developed to predict antigenic determinants on the proteins [35].

The conformational and linear epitopes’ predictions were performed based on the alignment of the sequences of different CHIKV isolates (access numbers: KP164572 (Asian), KJ796852 (ECSA-IOL), KY055011 (ECSA), and HM045817 (WA)) and using the IEDB’s program package by National Institute of Allergy and Infectious Diseases(Bethesda, MD, USA) (https://www.iedb.org/) (accessed on 1 October 2020) [11]. Conformational epitopes were obtained after the analysis of the structures modeled in the Discotope 2.0 prediction algorithm (Haste Andersen P, Bethesda, MD, USA) (http://tools.iedb.org/discotope) (accessed on 1 October 2020), which integrates the combination of two scores linearly; one based on the hydrophobicity/hydrophilicity scale and a score of the epitopes’ propensity, which is based on the calculation of the surface accessibility and residue contact area [11,17]. The linear epitopes were obtained after analysis of the structures modeled in the ElliPro prediction algorithm (Ponomarenko, Bethesda, MD, USA)(http://tools.iedb.org/ellipro/) (accessed on 1 October 2020) [11,17], which is based on the residue protrusion index. The Pymol (The PyMOL Molecular Graphics System, Version 1.8 (Warren Lyford DeLano, New York, NY, USA) developed by Schrödinger, LLC.) (https://pymol.org/2/) (accessed on 1 October 2020) and Jalview Version 2.11.2.4 (Andrew Waterhouse, Dundee, Scotland) (www.jalview.org) (accessed on 1 October 2020) software were used for editing and indicating the conformational and linear epitopes in the modeled proteins and the aligned sequences, respectively, for each of the CHIKV structural proteins. Using the results obtained in the combined linear and conformational epitopes’ predictions with the previously obtained information in the literature [12,19] and focusing on epitopes conserved by all genotypes, a panel of 11 linear peptides (P1–P9 corresponding to the E2 protein and P10–P11 corresponding to the E3 protein) were selected for synthesis (Proteimax, São Paulo, Brazil) and epitope validation was obtained by Peptide-based indirect ELISA.

### 2.4. Enzyme Immunoassay Qualitative and Quantitative for IgG Anti-CHIKV Antibodies

To identify and quantify positive samples, we used the protocol for the IgG antibody developed by EuroImmun (Euroimmun, Lübeck, Germany). In the first step of the qualitative assay, the samples, calibrator 2, and the positive and negative controls were diluted 1:101 using the sample buffer and added to the microplate wells covered with CHIKV antigens. In the first step of the quantitative assay, the samples, calibrators 1, 2, and 3, and the positive and negative controls were diluted 1:101 and added to the microplate wells covered with the same CHIKV antigens. The samples were added to the plate, which was incubated for 1 h at 37 °C. In the second step, the plate was washed 3 times with 300 µL of a wash solution in each well and, after washing, 100 μL of enzymatic conjugate (anti-human IgG marked with peroxidase) was added to each well, and the plate was incubated at room temperature (18 to 25 °C) for 30 min. In the third step, the plate was washed 3 times with 300 µL of a wash solution and, after washing, 100 µL of a substrate/chromogenic solution was added to each microplate well following incubation at room temperature (18 to 25 °C) for 15 min. In the fourth step, 100 µL of stop solution was added to each well of the microplate, and the optical density (O.D.) was measured in an Epoch Microplate Spectrophotometer (BioTek, Winooski, VT, USA). The results were interpreted following the manufacturer’s instructions.

### 2.5. Peptides Screening Using Peptide-Based Indirect ELISA

To identify the target epitopes of CHIKV-specific IgG antibodies, 18 sera samples from positive-IgG CHIKV patients, collected in the state of Sergipe in 2016, were characterized with a quantitative immunoenzyme assay developed by EuroImmun. The sera samples were analyzed by the Euroimunne test with a mean titration of 181 UR/mL and standard deviation of 0.1 UR/mL and subsequently used in the standardization assay. All samples used in this study were tested negative for the presence of CHIKV antibodies IgM by the EuroImmun CHIKV IgM qualitative test. For indirect ELISA standardization using synthetic peptides as antigens, we briefly performed the following: in the first step of the experiment, a pool of 18 Chikungunya positive IgG samples identified by the EuroImmun test was used, and a commercial negative IgG serum with an unspecific reaction for the Chikungunya virus and identified by the EuroImmun test was used as a negative control (Catalog number: S7023; Sigma Aldrich, St Luis, MO, USA). The peptides P1 to P9 corresponding to the E2 protein and the peptides P10 and P11 corresponding to the E3 protein were diluted in a filtered PBS buffer (1 M pH 7.4) at a final concentration of 200 µg/mL; then, serial dilutions of factor 100× to a concentration of 0.2 fg/mL were performed to obtain a dilution curve of the antigen concentration. The peptide dilutions were used to coat the polystyrene Nunc 96 microplate wells ELISA plate, 50 µL of solution per well were added, and the plates were incubated at room temperature until the wells dried. For background reference, peptide-free wells were used solely with the blocking solution. The blocking step was done with 5% albumin, 1% powdered milk solution, and 0.05% Tween20, diluted in PBS 1× (pH 7.4), followed by the addition of 50 µL per well and overnight incubation at 4 °C or 37 °C for two hours. After the blocking solution was removed, 50 µL of the diluted serum (1:100) was added to the PBS 1× solution (pH 7.4) and incubated for 1 h at 37 °C. Then, the wells were emptied, and the plate was washed 3 times with 200 µL per well of a wash solution. After the washing step, 50 µL of an enzymatic conjugate (Ig anti-human marked with peroxidase 1:12,000) was added to the plate wells and incubated for 1 h at 37 °C. The plate was washed 4 times with 200 µL of a wash solution after incubation, and 100 µL of a substrate/chromogenic solution was added to each microplate well, followed by incubation at room temperature (18 to 25 °C) for 15 min. In the last step, 100 µL of a stop solution was added to each well of the microplate, and then the photometric measurement at 450 nm and 620 nm was performed, and the results of optical density were analyzed. All trials were performed in sample technical triplicate and using individual patients’ sera.

### 2.6. Validation of the Peptides Using Peptide-Based Indirect ELISA

Based on the previous screening results, the 5 peptides (P1–P5) that demonstrated the potential to respond differentially were selected for individual patient sera analysis, including 10 Chikungunya virus-positive IgG sera and, as the negative control, 8 Chikungunya virus-negative IgG sera samples previously characterized by the EuroImmun test. The P1 to P5 peptides corresponding to the E2 protein were diluted in a filtered PBS buffer (1M pH 7.4) at a final concentration of 20 µg/mL followed by the addition of 50 µL of a solution per well to coat the polystyrene Nunc 96 microplate wells; the plates were incubated at room temperature until the wells dried. For background reference, peptide-free wells were used solely with the blocking solution. The blocking step was done with 5% albumin, 1% powdered milk solution, and 0.05% Tween20, diluted in PBS 1× (pH 7.4), followed by the addition of 50 µL per well and overnight incubation at 4 °C or 37 °C for two hours. After the blocking solution was removed, 50 µL of the diluted serum (1:100) was added to the PBS 1× solution (pH 7.4) and incubated for 1 h, at 37 °C. Then the wells were emptied, and the plate was washed 3 times with 200 µL per well of a wash solution. After the washing step, 50 µL of enzymatic conjugate (Ig anti-human marked with peroxidase 1:12,000) was added to the plate wells and incubated for 1 h at 37 °C. The plate was washed 4 times with 200 µL of a wash solution after incubation, and 100 µL of a substrate/chromogenic solution was added to each microplate well, followed by incubation at room temperature (18 to 25 °C) for 15 min. In the last step, 100 µL of a stop solution was added to each well of the microplate, and then the photometric measurement at 450 nm and 620 nm was performed, and the results of the optical density were analyzed. All trials were performed in sample technical triplicate and using individual patients’ sera.

### 2.7. Data Analysis

All experiments were performed in sample triplicate. All data were presented as means and standard deviations. The differences in responses between the analyzed group and the control group were analyzed using appropriate statistical tests. The Graphpad software was used for the analysis of classifier performance, and a Receiver Operating Characteristic curve (ROC curve) was performed for each one of the best peptides (P1–P5) with a 95% confidence interval; the sensitivity corresponds to the rate of true positives and the specificity to the rate of true negatives. Most of the graphs presented here were performed using R scripts and are available upon request.

## 3. Results

To select representative amino acid sequences from all genotypes and lineages spread worldwide causing the human CHIKV disease, we relied on phylogenetic reconstruction and tree topology, where four representative sequences were chosen for each of the Asian, ECSA, and WA genotypes and the IOL lineage (GenBank accession numbers: KP164572, KY055011, and HM045817 KJ796852) (Figure 1). Based on the obtained structures in the modeling step, the conformational and linear epitopes were predicted (Figure 2 and Figure 3, Appendix A).

Based on the prediction results, the E2 protein (Figure 2, Supplementary Appendix A) presented the highest immunogenic potential compared with the E3 protein (Figure 3, Appendix A) and demonstrated both conformational and linear epitopes in the conserved regions among the genotypes and/or lineages with both predictions demonstrating the E2 protein as the main target of the immune response (Figure 2, Appendix A). In the epitopes’ prediction for the E3 protein (Figure 3, Appendix A), only one potential epitope was identified at positions 57 to 61 in the C-terminal region in both the linear and conformational predictions. Using the results obtained in the combined linear and conformational predictions with the literature information previously obtained [12,19] and focusing on the epitopes preserved among all the genotypes and/or lineages, a panel of 11 linear peptides was selected for epitope validation. For validating the epitopes and checking the serological reactivity, a group of CHIKV-positive IgG patient samples was tested and selected (Table 1) for peptide validation. The 11 selected peptides were synthesized and visually located in the E2 and E3 proteins crystallography structure (Figure 4 and Figure 5) and then tested in the peptide-based indirect immunoenzyme assay.

The peptides P1 to P9 were synthesized with a size of 18 amino acids corresponding to protein E2 (Figure 4, Table 2), and the peptides P10 and P11 were synthesized with a size of 12 amino acids corresponding to the E3 protein (Figure 4, Table 2). The peptides P3, P4, P5, P8, and P9 are conserved among all the genotypes (Figure 4). The peptides P1, P6, and P10 correspond to the genotypes and/or lineage WA, ECSA, and IOL sequences (semi-conserved), while the P2 and P11 peptides are variants of these peptides, corresponding to the Asian genotype sequence exclusively, and P7 corresponding to the IOL genotype sequence exclusively (Figure 4, Table 2).

To identify the target epitopes of CHIKV-specific IgG antibodies, a pool of 18 positive patient sera (IgG + CHIKV) and a non-specific human IgG serum as the control were used to screen for the epitopes that best react to previously characterized antibodies. The results indicated that the maximum peak achieved at a 1 ng concentration per well was obtained as the best reactivity and also the best difference between CHIKV IgG positive and the human IgG non-specific serum (Figure 6). Later, this concentration was used as a standard in the following assays with individual samples. In the peptide standardization assay, the P4 and P5 peptides were found to be the most reactive, respectively, and were selected to be evaluated with the individual samples from the CHIKV IgG positive and CHIKV IgG negative patients (Figure 7, Table 1). The previous epitopes, as described by Kam et al. [13,26], and corresponding to peptides P1–P3, were also included for the analysis with the individual samples. In this analysis, 10 positive IgG CHIKV sera samples with a mean titration of 160 UR/mL and a standard deviation of 46.87 UR/mL, characterized by the EuroImmun assay, were used. As controls, eight negative IgG CHIKV sera samples were used (Figure 6).

The following cut-off value was established as the comparison threshold, the average of the triplicate two standard deviations (the mean plus two SDs). The peptides P1 and P2 did not obtain a reactivity above the established cut-off value and demonstrated no potential for discrimination among the groups evaluated (Figure 7, Table 1). The peptides P3, P4, and P5 reacted above the established cut-off value and showed a discriminant potential among the groups evaluated (Figure 7, Table 1). Peptides P3 and P4 showed a stronger reactivity in the recognition among the peptides that reacted above the cut-off value. With the results obtained in this analysis, the peptides P1 to P5 were also analyzed for sensitivity and specificity through the analysis of the ROC (Receiver Operating Characteristic curve) (a 95% confidence interval) to verify the potential for the correct discrimination among the analyzed sample groups (Figure 8, Table 1). The P1 and P2 peptides demonstrated a low potential for classification among the analyzed groups and generated an AUC (Area under the curve) of below 0.7. The P3, P4, and P5 peptides, in addition to being reactive, demonstrated potential as discriminator classifiers, with AUC values of 0.86, 0.99, and 0.9, respectively (Figure 8, Table 1).

## 4. Discussion

In general, the surface proteins of the viruses are targets for neutralizing antibodies. The E3 and E2 CHIKV proteins have the function of binding to cell receptors, and the E1 protein is the protein responsible for membrane fusion in the endosomal vesicle protected by the E2 glycoprotein and is structurally less accessible to B lymphocyte antibodies, consequently with limited and low immunogenic potential. Our study focused on predicting the B-cell epitopes of the CHIKV structural proteins, E2 and E3, verifying the reactivity of synthetic linear peptides corresponding to the epitopes mapped by utilizing the CHIKV IgG-positive serum samples collected in the state of Sergipe, Brazil, in 2016. Screening studies using different patient cohorts, different geographic regions, and different genetic backgrounds help to identify and define usable signatures for the development of serological assays and vaccines [12,13,19]. So far, there are no studies on the target of the immune response of the Chikungunya virus previously infecting patients in Brazil and South America.

E2EP3, corresponding to peptide P1, was described by Kam et al., 2012 [12,13,19] as a marker epitope of the early convalescent phase of the CHIKV infection studying a cohort of patients on the Asian continent. In this study, Kam et al., 2012 [12,13,19] suggest that an ELISA based on the epitope E2EP3 may be useful to study CHIKV infections and to determine the magnitude of outbreaks. An analysis of this epitope with the sampled patient cohort showed no strong reactivity and potential as a marker for the CHIKV infection. These results may be explained by not using an ELISA protocol employing detection amplifiers as performed in the study by Kam et al., 2012 [12,13,19]. Another reason might be the sample collection period is not nearly the early stage of convalescence, considering the response pattern varies over time, and the sampled groups under analysis have different genetic backgrounds [12,19]. The P2 peptide with two amino acid changes lost recognition by the serum from patients infected with the ECSA genotype (Figure 6). This difference in reactivity requires further investigation and preferably by comparing well-characterized convalescent-phase sera from patients infected with the different genotypes.

Other significant immunodominant epitopes found by Kam et al., 2012 [12,13,19] are located among the amino acids 3025–3058 (corresponding to peptides P5, P6, and P7) in the E2 protein between Domain B and C in a linker region. This region is associated with early protection from the convalescent to the recovery phase, being considered an epitope with potential vaccine application. In this study, Kam et al., 2012 [12,13,19] also concluded that a single substitution in the amino acid K252Q in glycoprotein E2 has an important effect in decreasing the antibody binding capacity. The P6 and P7 peptides are variants of the same peptide; P7 has a lysine at position 252, previously identified as an escape mutation and was poorly recognized by the pool of the sera in the standardization test compared to the P6 peptide according to Kam et al. from their 2012 [12,13,19] findings (Figure 5). Both P6 and P7 peptides are adjacent to the P5 peptide (Figure 4). The P5 peptide was the most reactive and was selected to be evaluated with the individual samples (Figure 6, Figure 7 and Figure 8, Table 1).

The peptides P7, P8, and P9 were weakly recognized by the specific CHIKV positive-IgG antibodies, similar to the findings described by Kam et al., 2012 [12,13,19] (Figure 6). These peptides correspond to the linker regions between Domains B and C (peptides P7 and P8), and after Domain C (peptide P9), they are accessible; however, they demonstrated a low potential for antigenic recognition (Figure 6; Appendix A). The peptides P10 and P11, corresponding to the C-terminal region of the E3 protein, were weakly recognized by the specific CHIKV IgG antibodies and demonstrated a low antigenic recognition potential (Figure 6; Appendix A).

The peptides P3, P4, and P5 are conserved among all the genotypes and are located in Domain A of the E2 protein (Figure 4, Table 2) and proved to be the most reactive peptides (Figure 6), showing the potential for antigenic recognition, being the major targets of the antibodies in the group of the patients sampled (Figure 6, Figure 7 and Figure 8, Table 1). The P3 peptide is adjacent to the epitopes described by Kam et al. in 2012 [12,13,19] (the P1 peptide), overlapping three amino acids and confirming the N-terminal region of the E2 protein as one of the major immunogenic targets of specific CHIKV IgG antibodies (Figure 2, Figure 6, Figure 7 and Figure 8; Appendix A). The P4 and P5 peptides are also located in immunogenic regions, as previously described, withP4 in Domain A and P5 at the end of Domain B, in a region of arc accessible to the antibodies between Domains B and C (Figure 2, Figure 3, Figure 5 and Figure 6; Appendix A). Additional peptides in Domain B would also be of interest to test. The peptides P1, P2, P3, and P4, corresponding to Domain A and peptide P5 at the end of Domain B, demonstrate this region as the main antigenic and immunogenic target of the E2 structural protein and are expected to be present in different antigen and vaccine designs [3,13,26].

Additionally, the peptides P4 and P5 are reactive and have the potential as markers for serological tests, demonstrated by the performance analysis of the classifiers observed in the ROC curve graphs (Figure 8; Table 1). The performance of these peptides requires further confirmation by analyzing a larger number of samples and, preferably, collected in different regions. This peptide panel is useful for identifying the target regions of the immune response in Chikungunya virus-infected patients. Finally, the most reactive peptides, P1, P3, P4, P5, and P6, were located in the Chikungunya virus’ glycoprotein envelope structural organization, resolved by x-ray crystallography (PDB:3N43), identifying the main antigenic and immunogenic epitopes analyzed in this study (Figure 9). A methodology with a higher detection capacity is also interesting and desirable, considering that the indirect peptide-based ELISA technique used in this study has limitations and is useful in exploratory studies. The potential biological activity of these peptides may also be analyzed in plate reduction neutralization assays to identify epitopes presenting a neutralizing potential, as well as for immunization with peptide-content vaccines [12,16,19]. Epitopes-based vaccines present an alternative capable of inducing a protective immune response with no adverse effects in vivo. Epitope mapping and validation studies are important for the discovery of B-cell epitopes that can be targeted by neutralizing antibodies and also for identifying T-cell epitopes that can be targeted by protective cytotoxic T-cells (CD8+ T-cells) and, finally, epitopes recognized by CD4 + auxiliary T-cells for the optimization of anti-CHIKV-specific antibody generation. The potential advantages of this strategy include safety, the ability to design and manufacture epitopes to increase vaccination efficiency, and the opportunity to design vaccines with a higher population coverage. Peptide vaccines containing sequences covering all genotypes have the potential to generate protection against CHIKV infections globally and, also, to cross-protect against the closely related Alphavirus [12,13,16,19].

## Figures and Tables

**Figure 1 viruses-14-01839-f001:**
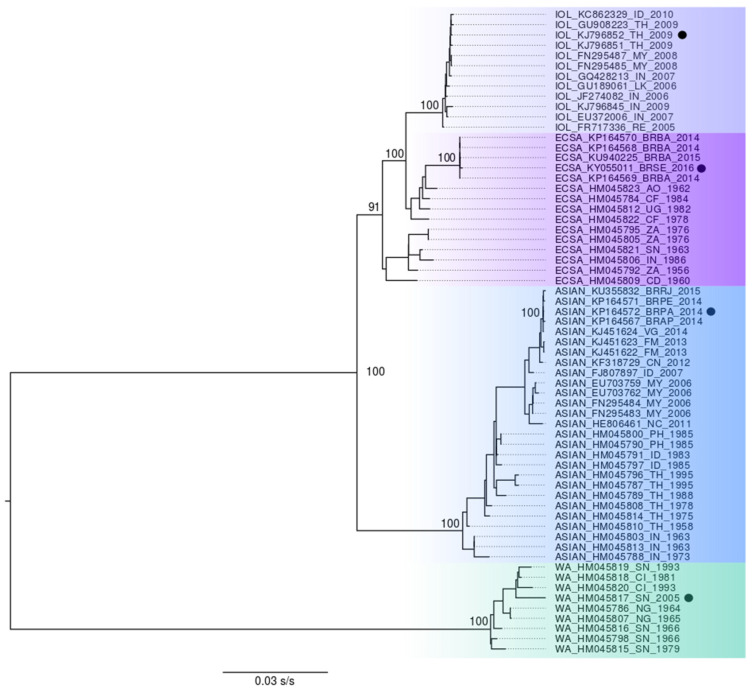
Phylogenetic tree representing the genotypes of the Chikungunya virus. The Chikungunya virus phylogenetic reconstruction, with 63 complete genome sequences. The green color is the WA genotype (West African), the blue color is the Asian genotype, the violet color is the ECSA genotype (East-Central-South African), and the gray color is the IOL lineage. The present values close to the main knots represent the bootstrap values.

**Figure 2 viruses-14-01839-f002:**
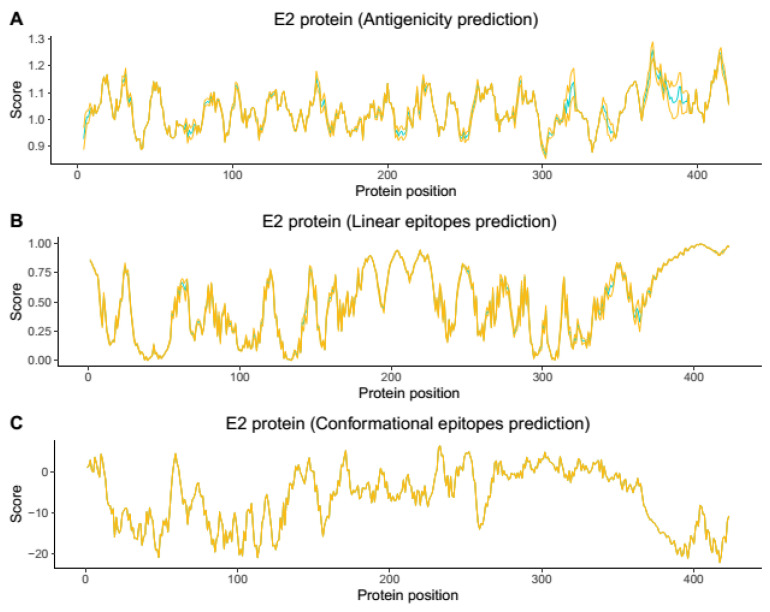
Chikungunya virus E2 Protein antigenicity (**A**) and linear (**B**) and conformational (**C**) epitopes’ prediction. The blue line represents the mean value, and the yellow line represents the standard deviation value. In general, the places where only yellow appears are regions with a very small standard deviation and probably conserved regions among the CHIKV genotypes, and where blue appears, they correspond to regions with a higher standard deviation and probably variable or semi-conserved regions among all four CHIKV genotypes.

**Figure 3 viruses-14-01839-f003:**
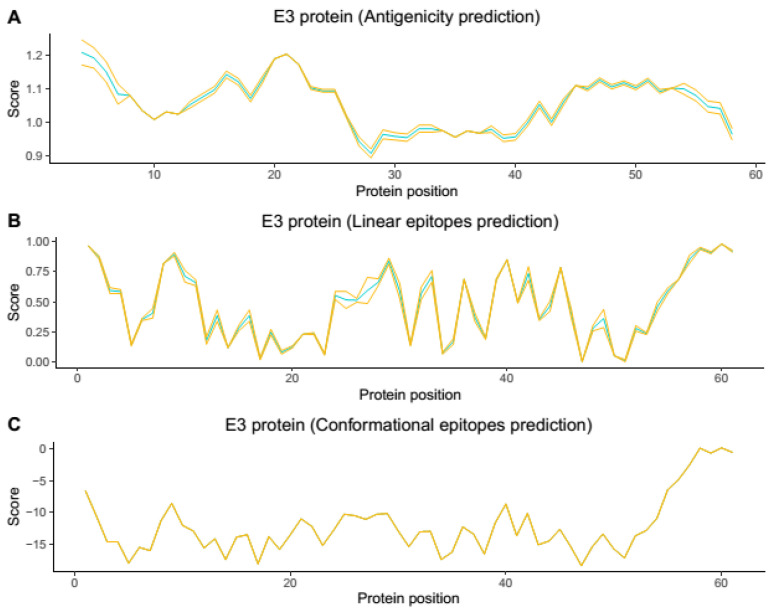
Chikungunya virus E3 Protein antigenicity (**A**) and linear (**B**) and conformational (**C**) epitopes’ prediction. The blue line represents the mean value, and the yellow line represents the standard deviation value. In general, the places where only yellow appears are regions with a very small standard deviation and probably conserved regions among the CHIKV genotypes, and where blue appears, they correspond to regions with a higher standard deviation and probably variable or semi-conserved regions among all four CHIKV genotypes.

**Figure 4 viruses-14-01839-f004:**
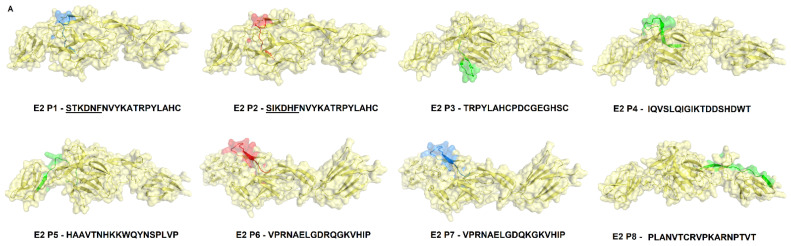
Synthetic peptides are highlighted on the E2 Protein structure and multiple sequence alignment. On top (**A**), predicted epitopes are highlighted on the structure of the protein E2. (**B**), Multiple alignment of E2 protein sequences corresponding to the four genotypes with highlighted peptides. In green, epitopes are conserved among all genotypes. In blue, epitopes are common to more than one genotype. In red are the single epitopes of each genotype. *—Amino acid changes in the regions predicted as epitopes. Underlined amino acids in peptides’ sequences corresponding to regions not resolved in the crystallography structure.

**Figure 5 viruses-14-01839-f005:**
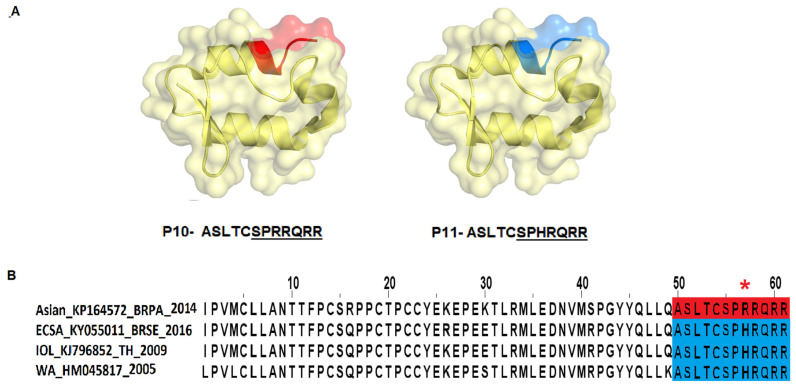
Synthetic peptides are highlighted on the E3 Protein structure and multiple sequence alignment. On top, (**A**) the predicted epitopes are highlighted on the structure of protein E3. (**B**), The multiple alignment of the E3 protein sequences corresponding to the four genotypes with highlighted peptides. Blue epitopes are common to more than one genotype. In red are the single epitope for individual genotype. *—Amino acid changes in the regions predicted as epitopes. Underlined amino acids in the peptides’ sequences corresponding to regions not resolved in the crystallography structure.

**Figure 6 viruses-14-01839-f006:**
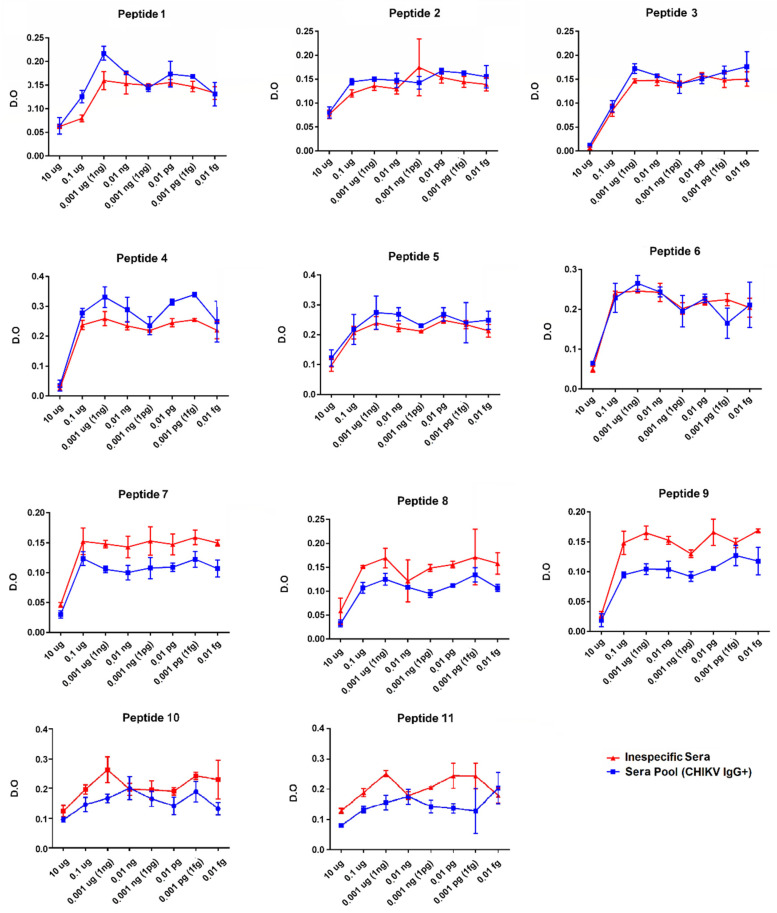
Standardization results of the indirect immunoenzyme peptide assay. A—The peptides P1–P9 correspond to the E2 protein. The peptides P10 and P11 correspond to the E3 protein. On the *Y*-axis, the optical density (O.D.) measurements for the positive IgG CHIKV patient sera pool (blue line) and, as the control, a non-specific IgG human serum (red line). On the X-axis the antigen dilutions. All assays were performed in sample triplicates.

**Figure 7 viruses-14-01839-f007:**
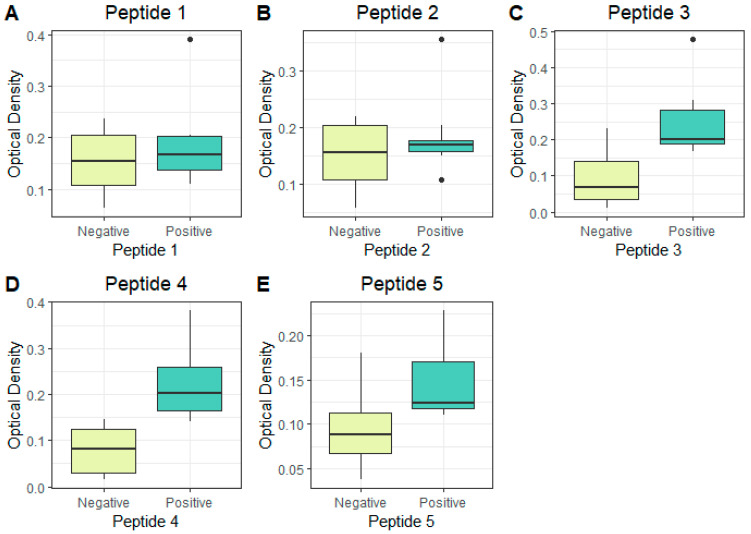
Protein E2 peptides’ P1 (**A**), P2 (**B**), P3 (**C**), P4 (**D**), and P5 (**E**) reactivity using individual sera samples. Boxplots in yellow represent the CHIKV IgG negative sera, and green boxplots represent the CHIKV positive-IgG patients. The experiments were all performed in sample triplicates. The following cut-off value was established as the comparison threshold; mean of the triplicates +2x the standard deviation.

**Figure 8 viruses-14-01839-f008:**
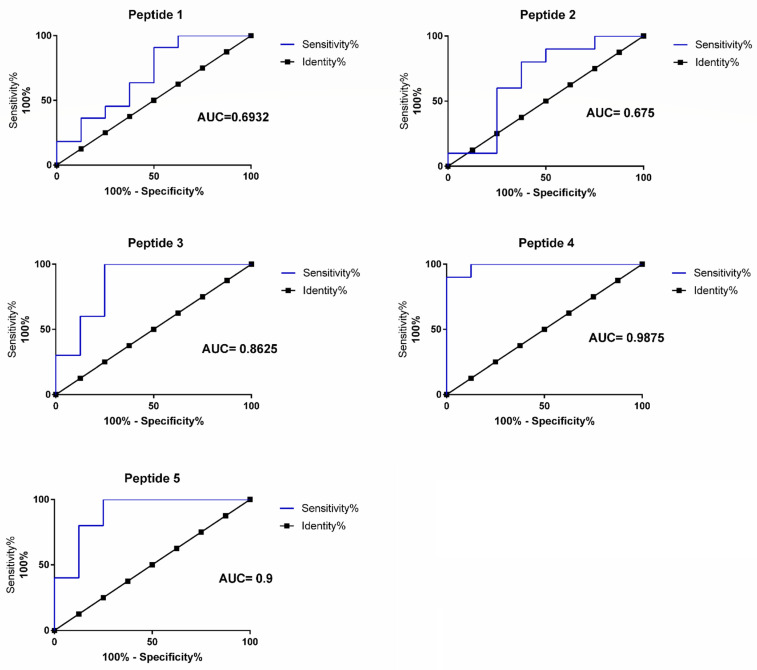
ROC curve for E2 protein synthetic peptides P1 to P5. 95% confidence interval. The peptides’, P1 to P5, potential as classifiers was analyzed by the Receiving Operating Curve analysis. On the x-axis specificity score and the y-axis sensitivity score, the black dotted line represents the identity score, and the blue line represents the sensitivity score. The AUC (Area Under the Curve) score is shown for the individual peptides.

**Figure 9 viruses-14-01839-f009:**
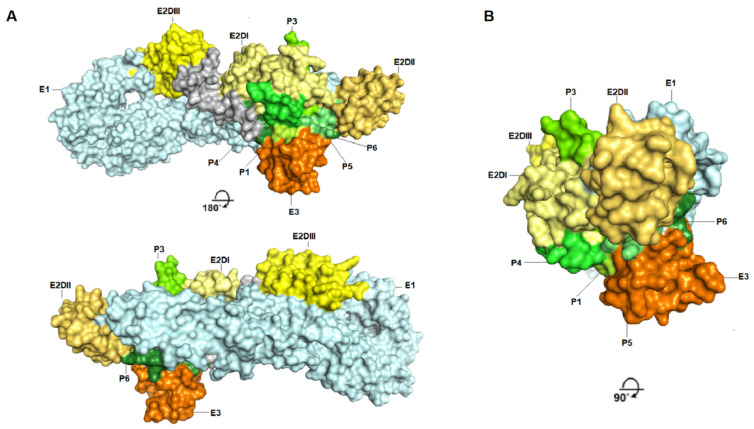
Antigenic and immunogenic B-cell epitopes are highlighted on the Chikungunya virus’ glycoprotein envelope structural organization. (**A**) The Chikungunya virus’ glycoprotein envelope complex structure’s surface (PDB:3N43); protein E1 is in light blue; protein E2, Domains I, II, and III are in pale yellow, yellow-orange, and light yellow, respectively; and protein E3 is in orange. Epitopes corresponding to peptides P1, P3, P4, P5, and P6 in protein E2 are highlighted in green tones. (**B**) The upper view from (**A**).

**Table 1 viruses-14-01839-t001:** Summary of E2 protein synthetic peptides’ P1–P5 reactivity analyzed by IgG positive CHIKV individual serum samples and the Area Under the Curve (AUC) analysis.

Peptides	Control Samples	D.O Mean Control Samples	D.O Mean Control Samples Standard Deviation	Threshold (Control Samples Mean + 2SD)	CHIKV IgG Positive Samples	D.O mean CHIKV IgG Positivie Samples	D.O Mean CHIKV IgG Positive Samples Standard Deviation	Area under the Curve (AUC)	AUC *p*-Value
**Peptide1**	**8**	0.147	0.057	0.259	**10**	0.186	0.079	0.6932	0.1604
**Peptide2**	**8**	0.141	0.055	0.251	**10**	0.182	0.065	0.675	0.2135
**Peptide3**	**8**	0.085	0.083	0.229	**10**	0.244	0.095	0.8625	0.0032
**Peptide4**	**8**	0.074	0.054	0.176	**10**	0.219	0.073	0.9875	0.0005
**Peptide5**	**8**	0.084	0.032	0.146	**10**	0.152	0.040	0.9	0.0045

**Table 2 viruses-14-01839-t002:** Synthetic linear oligopeptides corresponding to the immunogenic regions of the E3 and E2 structural proteins of the Chikungunya virus.

Protein	Position	Sequence	Peptide
**E3**	2788–2799	**ASLTCSPRRQRR**	P11
	2788–2799	**ASLTCSPHRQRR**	P10
**E2**	2800–2818	**STKDNFVYKATRPYLAHC**	P1
	2800–2818	**SIKDHFNVYKATRPYLAHC**	P2
	2812–2828	**TRPYLAHCPDCGEGHSC**	P3
	2848–2865	**IQVSLQIGIKTDDSHDWT**	P4
	3026–3043	**HAAVTNHKKWQYNSPLVP**	P5
	3042–3059	**VPRNAELGDRKGKIHIPF**	P6
	3042–3059	**VPRNAEFGDRQGKVHIPF**	P7
	3060–3077	**PLANVTCRVPKARNPTVT**	P8
	3190–3208	**CARRRCITPYELTPGATV**	P9

## Data Availability

Not applicable.

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
