# Peer review of "Chikungunya Virus E2 Structural Protein B-Cell Epitopes Analysis"

_viruses, 2022, doi:10.3390/v14081839_

Round 1

Reviewer 1 Report

The manuscript by da Cruz Silva and colleagues provides a primary analysis of the reactivity of serum samples against synthetic peptides mapping chikungunya virus E2 and E3 envelope proteins in an indirect IgG ELISA. Selection of peptides was based on bioinformatic prediction of antigenicity and of linear vs conformational epitopes. Sera were collected during a chikungunya virus outbreak in Brazil. ELISA screening with pooled sera first allowed discriminating the peptides specifically recognized by pools of chikungunya positive samples. Finally, by addressing a secondary analysis of individual samples the authors investigate the performance of the peptides in an ELISA assay.

Major limitations of the study are the number of samples analyzed and the narrow range of discrimination showed by the ELISA. On the other hand, the analysis of the cohort is new and suggests differences in the reactivity against peptides compared to previously analyzed cohorts in Singapore and Kuala Lumpur. 

Finally, in the manuscript version for revision Figure 4A and Table 2 are missing. 

Author Response

Reviewer #1

Reviewer #1: Major limitations of the study are the number of samples analyzed and the narrow range of discrimination showed by the ELISA. On the other hand, the analysis of the cohort is new and suggests differences in the reactivity against peptides compared to previously analyzed cohorts in Singapore and Kuala Lumpur. 

Reply: We are grateful for all the suggestions, which we accept and believe will improve the scientific work presented by our group. Yes, we describe only 10 samples. Despite being a low number of sample, they are representative of the Chikungunya virus outbreak in the period, which is confirmed by the epidemiological data in the reference 27.

Reviewer 2 Report

This manuscript by Cruz Silva et al. examines an IgG epitope recognition of CHIKV E2 and E3 linear synthetic peptides recognized using serum from patients in the convalescence phase of infection in Brazil. The authors generate 11 peptides corresponding to CHIKV E2 and E3 and test the recognition by serum. They conclude that the peptides P4 and P5 were the most reactive and specific among the 11 epitopes.

As the authors mentioned, similar studies have been done from different labs. Peptide P1, P6 and p7 were described by Kam et al 2012(reference: 12). The P3 peptide is adjacent to the epitopes described by same paper. Thus, I’m worried about the novelty of this manuscript. 

The following points should be addressed to strengthen the findings of the manuscript.

Major points:

1. Before using the serum to test the serological reactivity with the peptides, the convalescence serum should be characterized, such as neutralizing capacities against different CHIKV isolates, and recognition of CHIKV antigen.

2. To find the key amino acid in the novel peptide P3 and P4, alanine-scan analysis could be done.

3. I noticed that all the experiments were performed in sample triplicate. It would be better if the authors can have another biological repeat.

Minor comments

1. The Figure 4A is missing in this manuscript, which is supposed to show the predicted epitopes on protein E2.

2. Page 2 line 29, replace (iii) with (ii)

3. Page 8 line 19, why do the authors talk about table 2 first before table 1?

4. Figure 4B, it would be very helpful if the authors label the peptide number on the sequence.

5. Table 1. peptide number should be shown in this table

6. Page 13 line 2, do authors miss some sentences in the Figure 8 legend?

7. Figure 9, label peptide number and protein domain on the structure

Author Response

Article

Chikungunya Virus E2 structural protein B-cell epitopes analysis

Author’s reply to the reviewer's:

Reviewer #2

Reviewer #2 :1. Before using the serum to test the serological reactivity with the peptides, the convalescence serum should be characterized, such as neutralizing capacities against different CHIKV isolates, and recognition of CHIKV antigen.

Reply: We are grateful for all the suggestions, which we accept and believe can improve the scientific work presented by our group. It is important to note that the recognition of CHIKV antigen was performed by the quantitative immunoenzyme assay developed by EuroImmun described in methods section. At the end of the discussion session, as suggested, we introduce limitations, and also suggested both neutralization and immunization assays as well methodologies with higher detection capacity, which we believe could be enriching the scientific data presented in this article.

  1. To find the key amino acid in the novel peptide P3 and P4, alanine-scan analysis could be done.

Reply: We are grateful for the suggestion, which would undoubtedly add a lot to the information about the influence of key individual aminoacids in the binding activity and recognition by Chikungunya virus positive IgG samples. However, considering that our main objective is to present mainly information about (i) map the epitopes of CHIKV structural glycoproteins, E2 and E3 to identify specific IgG class antibody binding sites of CHIKV infected patients and identify the reactive epitopes of anti-CHIKV IgG antibodies induced by primary infection in humans infected with the CHIKV ECSA genotype; (ii) identify antigenic signatures that may be used in the development of serological diagnostic trials and/or vaccine development. We will carry out the alanine scan assay in future studies and hopefully with a extensively number of Chikungunya virus IgG samples immunologically characterized.

  1. I noticed that all the experiments were performed in sample triplicate. It would be better if the authors can have another biological repeat.

Reply: Our concern with data consistency was expressed by the care in our experimental design. All experiments, performed in triplicate, and with a group of samples from different experiments reflect this. Of course, a new repetition would reinforce our data, however, we have very little time to present them with the answers. We plan to continue this study, and we will consider this questioning in a new phase of the study, however, in its continuity.

Round 2

Reviewer 2 Report

I have no more further comments.